# Risk of Esophageal and Gastric Cancer in Patients with Type 2 Diabetes Receiving Glucagon-like Peptide-1 Receptor Agonists (GLP-1 RAs): A National Analysis

**DOI:** 10.3390/cancers16183224

**Published:** 2024-09-22

**Authors:** Mark Ayoub, Rafi Aibani, Tiana Dodd, Muhammed Ceesay, Muhammad Bhinder, Carol Faris, Nisar Amin, Ebubekir Daglilar

**Affiliations:** 1Department of Internal Medicine, Charleston Area Medical Center, West Virginia University, Charleston, WV 25304, USA; rafi.aibani@vandaliahealth.org (R.A.); tiana.dodd@vandaliahealth.org (T.D.); muhammed.ceesay@vandaliahealth.org (M.C.); muhammad.bhinder@vandaliahealth.org (M.B.); nisar.amin@vandaliahealth.org (N.A.); 2Department of Internal Medicine, Bayonne Medical Center, Bayonne, NJ 07002, USA; farisc@marshall.edu; 3Division of Gastroenterology and Hepatology, Charleston Area Medical Center, West Virginia University, Charleston, WV 25304, USA

**Keywords:** gastric cancer, esophageal cancer, genetic, diabetes, GLP-1 receptor agonists

## Abstract

**Simple Summary:**

We conducted a retrospective multicenter national study examining the risk of gastric and esophageal cancer in patients with type 2 diabetes who were receiving glucagon-like peptide receptor agonists. We used propensity score matching analysis to account for confounders. We assessed the risk of the outcomes in a seven-year timeframe. We add to the most recent literature findings that debunk the risk of cancers with their use.

**Abstract:**

Introduction: Glucagon-like peptide-1 receptor agonists (GLP-1 RAs) are becoming more popular in managing type 2 diabetes mellitus (T2DM). Concerns linger over potential links to malignancies like pancreatic and thyroid cancers, requiring more research to clarify their safety profiles. Additionally, evidence suggests GLP-1 RAs may lower colorectal and pancreatic cancer risk, especially in obese and overweight individuals, indicating a protective effect beyond weight loss. Current studies leave a gap in comprehensively understanding cancer risks associated with GLP-1 RAs, which prompts further research to enhance our understanding of their overall safety. Methods: We queried the US Collaborative Network (63 health care organizations) of the TriNetX research database. Patients with T2DM were identified and divided into two cohorts: patients on GLP-1 RAs and patients not on GLP-1 RAs. We excluded tobacco use and alcohol use disorders, obese patients with a body mass index (BMI) of >25 kg/m^2^, and those with a family history of gastrointestinal malignancy, infectious mononucleosis, chronic gastritis, pernicious anemia, helicobacter pylori infection, or gastroesophageal reflux disease (GERD). We used a 1:1 propensity score matching (PSM) model using patients’ baseline characteristics, medications, labs, and genetics. We compared the rate of gastric cancer and esophageal cancer at the seven-year mark. Results: A total of 2,748,431 patients with T2DM were identified. Of those, 6% (*n* = 167,077) were on a GLP-1 RA and 94% (*n* = 2,581,354) were not on a GLP-1 RA. After PSM, both cohorts included 146,277 patients. Patients with T2DM who were on a GLP-1 RA, compared to those who were not, had a statistically significant lower risk of both gastric cancer (0.05% vs. 0.13%, *p* < 0.0001) and esophageal cancer (0.04% vs. 0.13%, *p* < 0.0001) at the seven-year mark. Conclusion: The use of GLP-1 RAs in patients with T2DM does not significantly increase the risk of gastric or esophageal cancer. This finding supports the continued use of GLP-1 analogues as a therapeutic option in managing T2DM, considering their well-established benefits and low risk of complications. Based on the study results, these medications may even have a protective effect against these malignancies.

## 1. Introduction

GLP-1 receptor agonists, recognized for their efficacy in managing type 2 diabetes mellitus, include FDA-approved drugs like exenatide, liraglutide, and semaglutide, and are particularly valued for their potential to reduce cardiovascular risk, a leading cause of mortality in these patients. However, amidst their benefits, concerns have been raised about their association with malignant neoplasms, including pancreatic cancer and thyroid C-cell cancer, based on animal and human studies, although a definitive link has not yet been established. This ambiguity highlights the necessity for further research, emphasized by statements from regulatory bodies like the FDA and EMA. Existing research primarily focuses on pancreatic cancer, leaving a gap in understanding the comprehensive cancer risk associated with these drugs. The current meta-analysis aims to bridge this gap, examining the association between GLP-1 receptor agonists and a broader range of malignant neoplasms, thereby aiming to enhance the understanding of their safety profile and address the critical concerns related to cancer risks in patients with type 2 diabetes [1,2,3].

Studies of the glucagon-like peptide-1 receptor (GLP-1R) expression across various human tumors have found it predominantly in specific endocrine, embryonal, and brain tumors, but it is notably scarce or absent in most carcinomas [4,5]. Insulinomas, primarily benign pancreatic islet cell tumors causing severe hyperinsulinism, show the highest levels of GLP-1R expression, making them of special clinical interest for targeted diagnosis and potential treatment due to the limitations and risks associated with conventional imaging and diagnostic techniques [6]. Although other endocrine tumors like gastrinomas and extrapancreatic tumors, including ileal carcinoids and medullary thyroid carcinomas, also express GLP-1Rs, the expression is significantly lower compared to insulinomas. Moreover, embryonal and nervous system tumors show modest GLP-1R expression, whereas most carcinomas, including those commonly affecting the breast, colorectal system, gastric system, and lungs, exhibit minimal or no GLP-1R expression [5,7]. The association of GLP-1 RAs and various cancers, and the expression of GLP-1Rs in the gastric and esophageal tissues, raises concerns about their effect on those organs and if there is an association between their use and cancer development. The data on such associations is very scarce and warrants further evaluation.

## 2. Materials and Methods

### 2.1. Inclusion and Exclusion Criteria

Our study was approved by the Charleston Area Medical Center Institutional Review Board (IRB) Committee under number 24-1116. Written informed consent was not needed and was waived, as we used the TriNetX de-identified database. The TriNetX (Cambridge, MA, USA) database is a global federal research network that combines real-time data with electronic medical records. We conducted our study using the US Collaborative Network, which comprises 63 health care organizations (HCOs) from the United States of America [8]. We identified patients with type 2 diabetes mellitus in the US Collaborative Network and divided them into two groups: patients receiving GLP-1 RAs and those who were not. We excluded patients with tobacco use and alcohol use disorders, overweight patients, defined as having a body mass index (BMI) of >25 kg/m^2^, and those with a family history of gastrointestinal malignancy, infectious mononucleosis, chronic gastritis, pernicious anemia, helicobacter pylori infection, or gastroesophageal reflux disease (GERD). The previously mentioned inclusion and exclusion criteria are highlighted in the Appendix A using specific International Classification of Diseases (ICD) -10 codes and genetic codes. We compared the rates of gastric and esophageal cancer between the two cohorts over seven years.

### 2.2. Statistical Analysis

Patients with type 2 diabetes mellitus who were ≥18 years old were identified. Those who were included in our study were divided into two groups: The first group included patients with type 2 diabetes mellitus who were being treated with a GLP-1 RA, and the second group included patients with type 2 diabetes mellitus who were not being treated with a GLP-1 RA. After allocation, we performed propensity score matching (PSM) in both groups to ensure successful and effective balancing of the groups. This was conducted via TriNetX innate logistic regression analysis using patients’ baseline demographics, genetic components, lab values, and medications received. The demographics included in PSM included age at index, race, and gender. The genetic and lab components used were MLH1, MSH6 Variants, MSH2 Variants, STK11 Variants, TP53 Variants, EPCAM, PMS2, CDH1 Variants, BMPR1A, SMAD4, APC genes, and hemoglobin A1C (HbA1C) levels. The medications used in PSM were lansoprazole, esomeprazole, pantoprazole, omeprazole, dexlansoprazole, ranitidine, cimetidine, famotidine, bismuth subcitrate, and anti-diabetic medications including biguanides, sulfonylureas, alpha glucosidase inhibitors, thiazolidinediones, and sodium-glucose transporter 2 (SGLT-2) inhibitors. A full comparison before and after the matching of PSM components is shown in our outcomes.

An outcome statistical analysis was subsequently conducted using the TriNetX platform. To assess the association of GLP-1 RA use and risk of gastric and esophageal cancer between cohorts, we used cumulative incidence and log-rank tests. We also calculated the risk ratio (RR) for each cancer with its respective confidence interval (CI). We used a *p*-value < 0.05 to show statistically significant data. A study flow chart is shown in Figure 1.

## 3. Results

### 3.1. Baseline Characteristics

Approximately 2,748,431 patients with T2DM were noted in the US Collaborative Network Database; 6% (*n =* 167,077) were being treated with a GLP-1 RA, and 94% (*n =* 2,581,354) were not being treated with a GLP-1 RA. The GLP-1 RA group had a mean age of 58 with a standard deviation (SD) of 13.5, as compared to 61.6 with an SD of 16.4 in the non-GLP-1 RA group. Approximately 51.1% of the patients receiving a GLP-1 RA were females, as compared to 47.1% in the other group; 64.1% of the GLP-1 RA receivers were White, as compared to 55.5% in the non-receivers; and 15.7% of the GLP-1 RA group were Black compared to 15.2%, while 4.5% of the GLP-1 RA group were Asian compared to 6.1%. Of the GLP-1 RA group, 8.4% were on pantoprazole compared to 2.9%, 7.2% were on omeprazole compared to 2.2%, 1.8% were on esomeprazole compared to 0.7%, and 1% were on lansoprazole compared to 0.4%. Additionally, 6.9% of the GLP-1 RA group were on famotidine compared to 2.1%, and 2.2% were on ranitidine compared to 0.8%. In terms of anti-diabetic medications, of the GLP-1 RA group, 56.8% were on biguanides, 25.8% were on sulfonylureas, 0.4% were on alpha glucosidase inhibitors, 6.7% were on thiazolidinediones, and 18.1% were on SGLT-2 inhibitors. Of the non-GLP-1 RA group, 8.6% were on biguanides, 3.4% were on sulfonylureas, 0.01% were on alpha glucosidase inhibitors, 0.7% were on thiazolidinediones, and 0.9% were on SGLT-2 inhibitors. The mean HbA1C level in the GLP-1 RA group was 8.3, with a standard deviation of 2.2, compared to 7.2, with a standard deviation of 2, in the non-GLP-1 RA group.

### 3.2. Outcomes

Following PSM, our patient pool comprised 292,554 patients, which were split equally into two groups: one that was receiving a GLP-1 RA for T2DM and another for those who were not receiving a GLP-1 RA. There was no statistically significant difference between the two groups in the previously mentioned variables. In the GLP-1 RA group, the mean age was 58.3, with an SD of 13.5. Approximately 51.2% of the group were females. More than half the group was White 63.3%, while Blacks constituted 15.8% and Asians constituted 4.5%. In terms of medications, 8.3% were on pantoprazole, 7% were on omeprazole, 1.8% were on esomeprazole, and 1% were on lansoprazole. Additionally, 6.6% of the GLP-1 RA group were on famotidine, and 2.2% were on ranitidine. As for anti-diabetic medications, 51.7% were on biguanides, 21.2% were on sulfonylureas, 0.3% were on alpha glucosidase inhibitors, 5.1% were on thiazolidinediones, and 9.7% were on SGLT-2 inhibitors. The mean HbA1C level in the GLP-1 RA group was 8.2, with a standard deviation of 2.2. A full list of PSM components before and after match is shown below in Table 1.

#### 3.2.1. Risk of Gastric Cancer

After performing PSM, we compared the rate of gastric cancer between patients receiving a GLP-1 RA and those who were not at the seven-year margin.

After seven years of therapy, the gastric cancer risk between the two groups was found to be lower in the GLP-1 RA group. Patients on a GLP-1 RA had a statistically significant lower risk of 0.05% compared to 0.13% in patients who were not on a GLP-1 RA (*p* < 0.0001). The risk ratio of developing gastric cancer was 0.4, with a 95% confidence interval of (0.309, 0.521). The calculated gastric cancer risk reduction was 61.5% in seven years.

A summary including the log-rank test results and hazard ratios is highlighted below in Table 2.

#### 3.2.2. Risk of Esophageal Cancer

Similarly, after performing PSM, we compared the rate of esophageal cancer at the seven-year margin between patients receiving a GLP-1 RA and those who were not.

After seven years of therapy, the esophageal cancer risk between the two groups was found to be lower in the GLP-1 RA group. Patients on a GLP-1 RA had a statistically significant lower risk of 0.04% compared to 0.13% in patients who were not on a GLP-1 RA (*p* < 0.0001). The risk ratio of developing gastric cancer was 0.332, with a 95% confidence interval of (0.249, 0.441). The calculated gastric cancer risk reduction was 69.1% in seven years.

A summary including the log-rank test results and hazard ratios is highlighted below in Table 3.

A bar graph comparing each cancer risk between both cohorts is shown in Figure 2.

## 4. Discussion

### 4.1. GLP-1 Receptor Agonists: Indications, Benefits, and Concerns

GLP-1 agonists, also referred to as GLP-1 receptor agonists (GLP-1 RAs), incretin mimetics, or GLP-1 analogs, are a class of medications used to manage type 2 diabetes mellitus (T2DM) and, in certain cases, obesity. Drugs in this category include exenatide, liraglutide, dulaglutide, and semaglutide. The American Diabetes Association (ADA) identifies metformin as the first-line treatment for T2DM. Nonetheless, the addition of GLP-1 RAs is recommended for patients who cannot tolerate metformin, those with hemoglobin A1c levels exceeding the target by more than 1.5%, and those who fail to achieve their target A1c within three months. This is particularly pertinent for patients with atherosclerosis, heart failure, and chronic kidney disease [9,10,11,12]. Additionally, semaglutide and liraglutide are FDA-approved as obesity treatments and are being prescribed to overweight patients with comorbid conditions. Research indicates that GLP-1 RAs show promise in improving hemoglobin A1c levels and promoting weight loss in patients with type 1 diabetes mellitus (T1DM).

According to the 2023 ADA guidelines, GLP-1 RAs are recommended for reducing cardiovascular risk. These medications not only decrease the likelihood of cardiovascular events and hypoglycemia, but also show promise in slowing the progression of chronic kidney disease (CKD). GLP-1 RAs are particularly recommended for patients with a history of clinical atherosclerotic cardiovascular disease (ASCVD), such as previous myocardial infarction or stroke. GLP-1 RAs that have demonstrated cardiovascular benefits include liraglutide, subcutaneous semaglutide, and dulaglutide [13,14].

Large-scale studies have investigated the cardiovascular safety of GLP-1 RAs. The ELIXA study focused on patients with T2DM and established cardiovascular disease (CVD) who were administered either lixisenatide (a GLP-1 agonist) or a placebo in addition to standard care. The results showed that the addition of lixisenatide did not significantly impact the rate of major cardiovascular events [15]. In the LEADER trial, patients with T2DM and established cardiovascular disease (CVD) were given either liraglutide (a GLP-1 agonist) or a placebo in addition to standard care. The results demonstrated that adding liraglutide significantly reduced mortality from cardiovascular events compared to the placebo [16].

A randomized controlled trial demonstrated that GLP-1 mimics, such as exenatide and liraglutide, significantly promote weight loss in obese and overweight patients without diabetes [17]. Furthermore, in the treatment of T2DM in obese patients with severe glycosylated hemoglobin A1c (HbA1c) impairment, supplementing metformin with dulaglutide provided sustained benefits in glucose metabolism and body weight control via multiple mechanisms, including delayed gastric emptying, increased satiety, elevated resting energy expenditure, and direct effects on the brain’s appetite center [18,19].

The most common side effects of GLP-1 agonists include nausea, vomiting, and diarrhea, which can potentially lead to acute kidney injury due to volume contraction. Other side effects may include dizziness, mild tachycardia, infections, headaches, and dyspepsia. Additionally, injection-site pruritus and erythema are frequently observed, particularly with the longer-acting medications in this class [20,21,22]. Lastly, a meta-analysis has indicated a link between the use of GLP-1 RAs and an increased risk of gallbladder and biliary disorders, particularly with higher dosages and prolonged use [23,24,25,26,27].

To date, numerous epidemiological studies have established a positive correlation between type 2 diabetes mellitus (T2DM) and the incidence of various cancers, including endometrial, hepatobiliary, pancreatic, breast, prostate, and colorectal cancers [28]. Hyperinsulinemia in diabetic patients appears to be a primary factor contributing to an increased cancer risk. Elevated insulin levels reduce the concentrations of insulin-like growth factor (IGF) binding protein, which typically binds tightly to IGFs, resulting in higher levels of free IGF-1 in cells and tissues. Elevated IGF-1 levels have been associated with a heightened cancer risk, making patients with T2DM more prone to developing various malignancies compared to healthy individuals. The potential impact of GLP-1 receptor agonists on tumor development has only recently garnered attention [29,30].

The long-term effects of GLP-1 RAs on the thyroid gland are under investigation, as rodent studies have shown that liraglutide can stimulate calcitonin release, leading to C-cell hyperplasia and tumors in the thyroid gland. While the impact on humans is not yet clear, further research is necessary. As a result, GLP-1 RAs are not recommended for patients with a personal or family history of multiple endocrine neoplasia type 2A (MEN 2A), type 2B (MEN 2B), or medullary thyroid cancer [7,23,31,32].

Additionally, in patients using GLP-1 RAs, there have been reports of acute pancreatitis, including potentially fatal hemorrhagic and necrotizing types. The causal relationship between GLP-1 agonists and pancreatitis or pancreatic cancer remains uncertain [24]. Zhao et al., through rigorous scientific experiments, demonstrated that activating the GLP-1 receptor with liraglutide exerted an anti-tumor effect on human pancreatic cancer by inhibiting the PI3K/AKT pathway [30]. Moreover, a large-scale clinical study showed that GLP-1 analogs could lower mortality in patients with both prostate cancer and diabetes [30].

A meta-analysis of clinical studies revealed that treating patients who have type 2 diabetes mellitus and are obese with GLP-1 RAs did not increase the risk of breast cancers, acute pancreatitis, pancreatic cancer, or overall tumor neoplasia. Additional research has shown that GLP-1 RAs can inhibit prostate cancer growth by targeting the PI3K/AKT/mTOR and ERK/MAPK pathways and can also suppress pancreatic and prostate cancer cell growth by inhibiting the NF-kB pathway. While no conclusive clinical evidence has been found to suggest that GLP-1 RAs are tumorigenic, numerous studies have indicated their potential to inhibit the growth of ovarian, breast, prostate, and pancreatic cancers [29,30,31,32,33,34,35,36].

### 4.2. Esophageal Cancer: Epidemiology, Incidence and Prevalence, Risk Factors, and Genetic Predisposition

Esophageal cancer is the eighth most common cancer worldwide and the sixth leading cause of cancer-related mortality [37]. It is often diagnosed at an advanced stage due to its asymptomatic nature in the early phases, contributing to its poor prognosis [38]. In 2020, there were approximately 600,000 new cases of esophageal cancer, and about 540,000 deaths occurred as a result of this disease [39].

The global incidence of esophageal cancer varies significantly by geographic region, with higher rates observed in Asia, particularly in China, and parts of Africa and South America [40]. The age-standardized incidence rate (ASIR) is notably high in East Asia, with China accounting for more than half of all new cases globally [41]. In contrast, Western countries like the United States and many European nations report lower incidence rates, although these rates have been rising in recent decades [37].

In 2020, the age-standardized rate of esophageal cancer was 6.3 per 100,000 people [42]. Although the age-standardized incidence rate decreased by 16.8% from 1990 to 2020, the total global incidence nearly doubled, increasing by 94.7% from 310,236 to 604,100 cases [37]. There is a notable gender disparity, with 418,350 cases in males and 185,750 in females [43]. Approximately 70% of new cases are in men, and the incidence of esophageal cancer is two to three times higher in men compared to women worldwide [44].

Some of the risk factors for esophageal cancer development include:

Smoking: Smoking is a significant risk factor for both Barrett’s esophagus and esophageal adenocarcinoma. Current smokers have nearly twice the risk of esophageal adenocarcinoma compared to nonsmokers, with a higher risk in men [45]. Even after 10 years of cessation, former smokers remain at increased risk. For ESCC, the risk increases with the number of packs smoked per year, particularly for those smoking over 30 packs annually [38]. Alcohol Consumption: Alcohol metabolism produces acetaldehyde, which can cause DNA mutations, thus increasing ESCC risk, especially with high weekly alcohol intake [44]. However, alcohol consumption does not significantly affect esophageal adenocarcinoma incidence [44]. Gastroesophageal Reflux Disease (GERD): GERD significantly raises the risk of Barrett’s esophagus and esophageal adenocarcinoma. About 10% of GERD patients develop Barrett’s esophagus, and those with frequent heartburn or regurgitation have a fivefold higher risk of esophageal adenocarcinoma [39]. Obesity and Body Composition: Obesity and high BMI are risk factors for esophageal adenocarcinoma, with greater BMI levels correlating with higher risk [43]. However, high BMI levels are associated with a decreased risk of ESCC, although higher blood pressure correlates with increased ESCC risk [44]. Alcohol consumption, contributing to both hypertension and ESCC, serves as a confounding factor [41]. Environmental Exposures: Exposure to certain chemicals and toxins, such as polycyclic aromatic hydrocarbons and human papillomavirus (HPV) infection, are associated with a higher incidence of ESCC [38]. Socioeconomic Status: Lower socioeconomic status, which often correlates with poor diet, smoking, and alcohol use, is a significant risk factor [46].

Genetic factors are thought to play a role in the susceptibility to esophageal cancer. Barrett’s esophagus itself has a genetic component, and those with a family history of Barrett’s esophagus or EAC are at a higher risk [43]. The interplay between genetic susceptibility and environmental factors, such as diet, lifestyle, and infections, plays a crucial role in the pathogenesis of esophageal cancer [41,47].

### 4.3. Gastric Cancer: Epidemiology, Incidence and Prevalence, Risk Factors, and Genetic Predisposition

Gastric cancer remains one of the leading causes of cancer-related mortality worldwide. By 2040, the global burden of gastric cancer is expected to increase by 62 percent [48]. This is particularly staggering given that in 2022 alone, there were 968,784 reported cases of gastric cancer, with a mortality rate closely following at 660,175. Gastric cancer was the fifth leading cause of cancer-related death worldwide in 2022 [49]. The incidence is twice as high in men than in women, and risk factors vary by the anatomical location of the disease. For example, non-cardia gastric cancer contributes approximately to 80% of gastric tumors globally and has been linked to factors such as *H. pylori* infection, alcohol consumption, and high salt intake [50]. *H. pylori* has been shown to be the strongest known risk factor for gastric cancer [51]. On the other hand, proximal gastric cancer (cardia) is associated with obesity and GERD and is more prevalent in Western Europe and North America [50].

Due to the significant link between *H. pylori* infection and gastric cancer, numerous studies have been conducted in endemic regions to explore the eradication of this bacterium. Most notably, there was a large-scale eradication trial on the Matsu Islands in Taiwan that involved a comprehensive screening and eradication of *H. pylori* from 2004–2018 that resulted in a decrease in the prevalence of gastric cancer from 64.2% to 15.0% [52]. The mechanism by which *H. pylori* infection leads to gastric cancer has been thoroughly investigated. The bacterium possesses specific virulence factors such as cytotoxin-associated gene A, vacuolating cytotoxin A, and outer membrane proteins that activate cellular proliferation signaling pathways [53]. Screening and treating individuals for *H. pylori* have shown promising outcomes. The crude global prevalence of *H. pylori* in adults decreased from 52.5% before 1990 to 43.9% during 2015–2022 [54].

Obesity is a multifaceted threat to our global health that extends beyond its impact on cardiovascular health and metabolic disorders. One of the most concerning aspects is the correlation with gastric cancer, raising the urgency of addressing this global pandemic. The World Health Organization’s data from 2022 revealed that one in eight individuals worldwide are living with obesity, and that number has more than doubled since 1990 [55]. The proposed hypotheses underlining excessive adiposity and gastrointestinal cancer risks include altered insulin and IGF-1 signaling, chronic low-grade inflammation associated with obesity, and disruptions in sex hormone metabolism. A comprehensive cohort study examining BMI in early and middle adulthood linked overweight and obese BMI in early and middle adulthood to heightened risks of colorectal and non-colorectal cancers, including gastric, esophageal, liver, and pancreatic malignancies [56]. One meta-analysis aimed to look at gastric cancer specifically and found that increased BMI was associated with gastric cardia cancer [57]. Amidst these staggering statistics and findings, there is hope for the emergence of new weight loss therapies such as GLP-1s. The FDA has approved GLP-1s for weight loss in individuals with and without diabetes and has had promising results.

It is believed that 1–3% of the global burden of gastric disease is truly hereditary. Familial gastric cancer syndromes are classified as either hereditary gastric cancer polyps or hereditary gastric cancer without polyps. The three main syndromes involved include hereditary diffuse gastric cancer (HDGC), gastric adenocarcinoma and proximal polyposis of the stomach (GAPPS), and familial intestinal gastric cancer (FIGC). Hereditary diffuse gastric cancer is autosomal dominant that alters the E-cadherin genes (CDH1) and beta-catenin (CTNNA1). CDH1 variant carriers with confirmed HDGC should undergo prophylactic total gastrectomy, and individuals who defer should undergo yearly screening with endoscopy [58,59]. Gastric adenocarcinoma and proximal polyposis of the stomach is a rare gastric polyposis syndrome. It is autosomal dominant with incomplete penetrance with a significant predisposition for the development of gastric adenocarcinoma [60]. The pathogenic variants seen in GAPPS are found mostly in the APC gene [61]. Unlike HDGC and GAPPS, familial intestinal gastric cancer remains genetically unexplained. FIGC has been characterized as having an autosomal dominant inheritance pattern without gastric polyposis. In countries with high incidence rates (Japan, Portugal), diagnostic criteria that are analogous to the Amsterdam criteria for hereditary non-polyposis are used [62]. Furthermore, other hereditary cancer syndromes are associated with gastric cancer. Hereditary Non-polyposis Colorectal Cancer (HNPCC), also known as Lynch syndrome, has a known association with colon cancer, but also increases the risk of gastric cancer [61]. The pathogenic variants found in HNPCC are MLH1, MSH2, MSH6, PMS2, and EPCAM [61]. Two more hereditary syndromes, Peutz–Jeghers and Juvenile Polyposis syndromes, are associated with gastric cancer development. The pathogenic genetic variants seen in those syndromes are found in the STK11, SMAD4, and BMPR1A genes [61]. Overall, familial gastric cancer is thought to account for 10% of cases, with only 1–3% linked to a gene defect [63].

### 4.4. Our Study Strengths and Limitations

We incorporated specific inclusion criteria, exclusion criteria, and PSM components. We aimed to exclude modifiable risk factors for gastric and esophageal cancers while balancing the cohorts using non-modifiable ones. Our exclusion criteria of overweight defined as BMI > 25 kg/m^2^ ensures that the GLP-1 RAs were prescribed for the sole indication of T2DM and not for weight loss. Excluding tobacco use and alcohol use disorders was aimed at excluding them as risk factors for our studied cancers or other cancers that may be discovered earlier, which may alter our final cohort. This was done to make sure there was a clean relationship between our medication of study and our outcomes. Furthermore, in PSM, we used the previously mentioned genes that are associated with hereditary cancer syndromes to account for non-modifiable risk factors.

Our study findings are in line with the most recent clinical studies examining the safety of GLP-1 RAs. Recent studies show a lower risk of certain obesity-associated cancers when compared to metformin and insulin over the span of 15 years [64]. A very recently published study in the Journal of American Medical Association (JAMA) studied the effect of GLP-1 RAs and 13 obesity-related cancers in patients with T2DM [64]. They identified esophageal cancer as one of those obesity-related cancers and found a significantly lower risk of esophageal cancer in patients on GLP-1 RAs (HR, 0.60; 95% CI, 0.42–0.86) compared to ours (HR, 0.34; 95% CI, 0.26–0.45). Two more studies found a risk reduction in colorectal cancer [1] and pancreatic cancer [65], respectively. Another recent study similarly used the TriNetX database for their analysis and found a reduced colon cancer risk in patients with T2DM with or without overweight, however, with a more profound effect in overweight patients [1]. The other study also found a protective effect from pancreatic cancer in patients with T2DM [65]. Those studies used similar methodologies and had similar outcomes when studying GLP-1 RA and cancer association. The ability to reproduce findings consistent with the recently published available studies supports the accuracy of the findings. Additionally, the size of our patient cohort from a nationwide multi-institutional database allows for the generalizability of the findings by increasing the statistical power of our analysis. Additionally, the use of PSM mitigates additional selection bias.

Our study has some limitations. First, this is a retrospective study of the de-identified database, which has inherent limitations including misdiagnosis and uncontrolled confounders. Despite our attempts to mitigate such limitations by using PSM for a detailed list of variables, these limitations and biases could not be fully eliminated. We were unable to account for some of the modifiable risk factors that are known to play a part in cancer development such as diet and physical activity due to the retrospective nature of the study. Second, our database is U.S.-based, which may affect its applicability outside of the USA. Lastly, due to the de-identified nature of the TriNetX database and the innate algorithm for the statistical analysis used within TriNetX, we were unable to confirm the duration of therapy or compliance with treatment. Therefore, future prospective studies are warranted to validate our findings.

## 5. Conclusions

In patients with type 2 diabetes mellitus, the use of GLP-1 RAs does not significantly increase the risk of gastric or esophageal cancer over the course of seven years. This finding supports and encourages the continued use of GLP-1 RAs as a beneficial therapeutic agent in managing patients with type 2 diabetes mellitus, considering their well-established benefits and low risk of complications. Based on the study results, these medications may even have a protective effect against these malignancies.

## Figures and Tables

**Figure 1 cancers-16-03224-f001:**
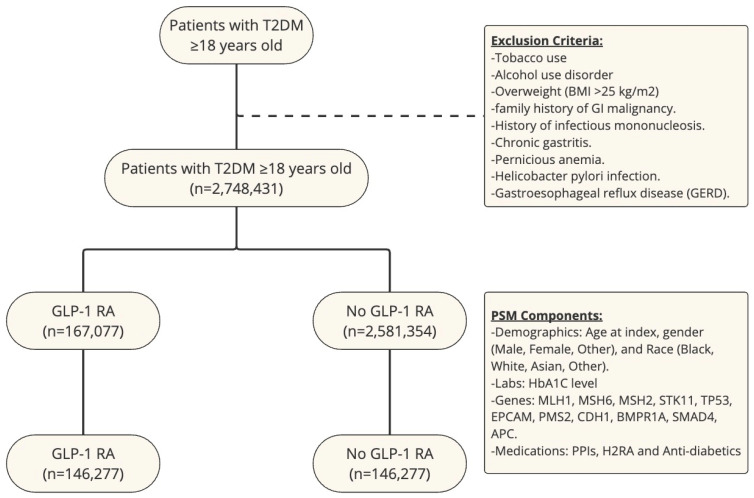
Study flow chart with exclusion criteria and PSM components.

**Figure 2 cancers-16-03224-f002:**
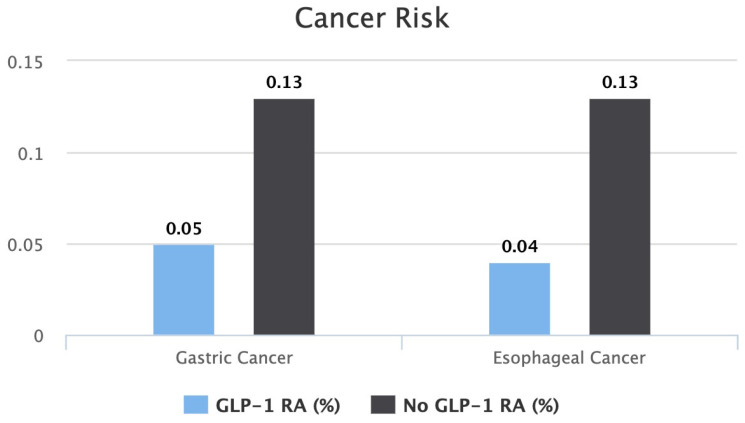
Bar graph comparing risk of gastric cancer and risk of esophageal cancer between patients on GLP-1 RAs vs. patients not on GLP-1 RAs.

**Table 1 cancers-16-03224-t001:** Comparison between the two cohorts before and after PSM.

	Before PSM			After PSM		
Characteristic	GLP-1 RA (*n* = 167,077)	No GLP-1 RA (*n* = 2,581,354)	*p*-Value	GLP-1 RA (*n* = 146,277)	No GLP-1 RA (*n* = 146,277)	*p*-Value
Demographics						
Age (Mean ± SD)	58.2 ± 13.5	61.6 ± 16.4	<0.001	58.3 ± 13.5	58.3 ± 14.4	0.995
White	64.1%	55.5%	<0.001	63.3%	63.9%	0.293
Black or African American	15.7%	15.2%	<0.001	15.8%	15.7%	0.385
Asian	4.5%	6.1%	<0.001	4.5%	4.3%	0.482
Female	51.1%	47.1%	<0.001	51.2%	51.4%	0.207
Medications						
Rabeprazole	0.09%	0.04%	<0.001	0.1%	0.1%	0.699
Lansoprazole	1%	0.4%	<0.001	1%	0.8%	0.083
Esomeprazole	1.8%	0.7%	<0.001	1.8%	1.6%	0.144
Pantoprazole	8.4%	2.9%	<0.001	8.3%	7.7%	0.174
Omeprazole	7.2%	2.2%	<0.001	7%	6.8%	0.430
Dexlansoprazole	0.3%	0.1%	<0.001	0.3%	0.2%	0.811
Ranitidine	2.2%	0.8%	<0.001	2.2%	1.7%	0.278
Cimetidine	0.1%	0.03%	<0.001	0.1%	0.1%	0.082
Nizatidine	0.0004%	0.006%	<0.001	0.01%	0.01%	0.746
Famotidine	6.9%	2.1%	<0.001	6.6%	6.2%	0.743
Bismuth subcitrate	0.02%	0.003%	<0.001	0.01%	0.01%	0.771
Biguanides	56.8%	8.6%	<0.001	51.7%	50.2%	0.092
Sulfonylureas	25.8%	3.4%	<0.001	21.2%	20.2%	0.154
Alpha glucosidase inhibitors	0.4%	0.01%	<0.001	0.3%	0.2%	0.761
Thiazolidinediones	6.7%	0.7%	<0.001	5.1%	5.3%	0.078
SGLT-2 inhibitors	18.1%	0.9%	<0.001	9.7%	11.6%	0.091
Labs and Genetics *						
HbA1C (Mean ± SD)	8.3 ± 2.2	7.2 ± 2	<0.001	8.2 ± 2.2	7.5 ± 2.2	0.084
PMS2 Variants	0.007%	0.004%	<0.001	0.007%	0.007%	0.097
MLH1	0.009%	0.007%	<0.001	0.009%	0.009%	0.102
MSH6 Variants	0.007%	0.003%	<0.001	0.007%	0.007%	0.405
MSH2 Variants	0.005%	0.002%	<0.001	0.005%	0.002%	0.405
STK11 Variants	0.004%	0.002%	<0.001	0.004%	0.004%	0.239
TP53 Variants	0.01%	0.005%	<0.001	0.01%	0.01%	0.125
EPCAM	0.007%	0.003%	<0.001	0.007%	0.007%	0.204
PMS2	0.007%	0.004%	<0.001	0.007%	0.007%	0.113
CDH1 Variants	0.002%	0.001%	0.009	0.002%	0.002%	0.502
BMPR1A	0.002%	0.001%	0.003	0.002%	0.002%	0.317
SMAD4	0.01%	0.005%	0.009	0.01%	0.01%	0.103
APC	0.017%	0.007%	<0.001	0.017%	0.017%	0.091

* For genetics, we used the closest 0.000 closest decimal due to the very small number of carriers of each gene.

**Table 2 cancers-16-03224-t002:** Comparison of rate of gastric cancer, log-rank test, and hazard ratio.

Outcome Gastric Cancer	GLP-1 RA (*n* = 146,277)		NO GLP-1 RA (*n* = 146,277)			*p*-value	
	0.05% (*n =* 79)		0.13% (*n =* 197)			<0.0001	
Log-Rank Test	X^2^		df			*p*	
	45.626		1			0.000	
Hazard Ratio and Proportionality	HR	95% CI		X^2^	df		*p*
	0.417	(0.321, 0.542)		7.370	1		0.001

**Table 3 cancers-16-03224-t003:** Comparison of rate of esophageal cancer, log-rank test, and hazard ratio.

Outcome Esophageal Cancer	GLP-1 RA (*n* = 146,277)		NO GLP-1 RA (*n* = 146,277)			* p *-value	
	0.04% (*n =* 63)		0.13% (*n =* 190)			<0.0001	
Log-Rank Test	X^2^		df			*p*	
	59.747		1			0.000	
Hazard Ratio and Proportionality	HR	95% CI		X^2^	df		*p*
	0.341	(0.257, 0.454)		1.409	0.341		0.002

## Data Availability

Available data is presented within the paper. Additional data is only available as permitted by third party.

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
