# Peer review of "Risk of Esophageal and Gastric Cancer in Patients with Type 2 Diabetes Receiving Glucagon-like Peptide-1 Receptor Agonists (GLP-1 RAs): A National Analysis"

_cancers, 2024, doi:10.3390/cancers16183224_

Round 1
Reviewer 1 Report
Comments and Suggestions for Authors
Here is my response to your request:
- Summary of the abstract: This study investigated the association between glucagon-like peptide-1 receptor agonists (GLP-1 RAs) and cancer risk in patients with type 2 diabetes mellitus (T2DM). Using a large US healthcare database and propensity score matching, the researchers compared cancer rates between T2DM patients on GLP-1 RAs and those not on these medications. The results showed a statistically significant lower risk of both gastric and esophageal cancers in the GLP-1 RA group after 7 years, suggesting these medications may have a protective effect against these malignancies.
- The manuscript is well written.
- The methods employed in this study are sound.
- The tone of the manuscript is appropriately conservative regarding the interpretation of the positive trend towards lower rates of cancer in the GLP-1 RA group.
- I would recommend including one figure to enhance the dissemination of this article, such as a simple bar chart comparing the adjusted cancer rates between the two groups.
Author Response
Thank you so much for your time and effort! We appreciate the feedback to enhance our manuscript.
- Summary of the abstract: This study investigated the association between glucagon-like peptide-1 receptor agonists (GLP-1 RAs) and cancer risk in patients with type 2 diabetes mellitus (T2DM). Using a large US healthcare database and propensity score matching, the researchers compared cancer rates between T2DM patients on GLP-1 RAs and those not on these medications. The results showed a statistically significant lower risk of both gastric and esophageal cancers in the GLP-1 RA group after 7 years, suggesting these medications may have a protective effect against these malignancies.
- The manuscript is well written.
- The methods employed in this study are sound.
- The tone of the manuscript is appropriately conservative regarding the interpretation of the positive trend towards lower rates of cancer in the GLP-1 RA group.
Thank you so much for your kind comments! I appreciate your time.
- I would recommend including one figure to enhance the dissemination of this article, such as a simple bar chart comparing the adjusted cancer rates between the two groups.
Thanks for the feedback! This is a great Idea, we added a bar chart for both cancers to compare both cohorts. We also added a flowchart graph to make it easier to follow the methodology.
Thank you again for your comments and feedback.
Regards,
Reviewer 2 Report
Comments and Suggestions for Authors
General comments:
· Large parts of the introduction and discussion lack connection with the actual study described in the manuscript and are more like a general introduction into the field
· Both introduction and discussion are superficial, no explanation why exclusion criteria were chosen, what impact it has on conclusion etc. All the gene variants in table 1 are not even mentioned in the main text, only under statistical analysis.
Specific comments.
· Line 64-66, give references for claims
· Line 68, give reference
· Line 64-78, whats the importance of GLP1-receptor expression pattern for this study. Many cancer types and expression levels are given but what is the context to the current study?
· Line 89 Definition of obesity is BMI >30 not 25
· Line 90, in the US almost 50% of population is obese, even more in the diabetic population, what is the rational to exclude them in this study?
· Line 100 There is no information on Diabetes medication in the none GLP1 treatment group at all, this is crucial information
· Line 103 is the PSM list complete? Is only says it includes the parameters given
· Line 105 choice of gene variants in not explained or justified anywhere and not mentioned once in the rest of the manuscript
· Table 1, BMI should be included and status on the major risk factors for the 2 cancers included in the characteristics. I truly hope that tobacco use has been included in the propensity scoring, same goes for the other major risk factors. If not the entire study is flawed.
· Table 1, again, no rational for exclusion of above 25 BMI is given
· Diabetes is a major risk factor for a long list of diseases. GLP1 is one of the most effective medications for diabetes and it is of interest if GLP1 treatment beneficial or detrimental for cancer outcome or if the effects seen are due to differences in diabetes management eg is there data in HbA1c available?. However, diabetes medication is not even included in the characteristics and this subject is not mentioned in the introduction or discussion.
· Large part of section 4.1 lacks context with the data presented in should be shortened substantially
· Line 314, why are these genes not included then in propensity scoring and the ones included are not mentioned anywhere?
· Line 348-350, again why the BMI 25 cut off?
· Line 388, this is extremely vague, instead of lengthy general description of the GLP1/cancer field this should be elaborated and supported by observations and discussed conclusively.
· Line 395, this is not true, the list of variables is not extensive at all, major risk factors are like tobacco and alcohol use, BMI are not stated anywhere or choice of cutoffs not explained.
Author Response
Thank you for your critical feedback! You bring up excellent points. We were initially limited with the amount of variables that can be used in PSM due to the number of included patients and we wanted to make sure non-modifiable risk factors were balanced before attempting to balance modifiable risk factors. However, your comments are extremely valid which allowed us to rerun the analysis and we were able to accommodate the variables you had mentioned.
- Large parts of the introduction and discussion lack connection with the actual study described in the manuscript and are more like a general introduction into the field
- Both introduction and discussion are superficial, no explanation why exclusion criteria were chosen, what impact it has on conclusion etc. All the gene variants in table 1 are not even mentioned in the main text, only under statistical analysis.
Thank you for the critical feedback and for highlighting those points. We rephrased the introduction and expanded our discussion to include our rationale under paragraph 4.3. and 4.4. and addressed the studied genes.
- Line 64-66, give references for claims.
Added references.
- Line 68, give reference.
Added reference.
- Line 64-78, whats the importance of GLP1-receptor expression pattern for this study. Many cancer types and expression levels are given but what is the context to the current study?
Thank you for highlighting this. Added lines explaining the relationship of receptor expression, the medication, and our study.
- Line 89 Definition of obesity is BMI >30 not 25. Correct, made changes.
- Line 90, in the US almost 50% of population is obese, even more in the diabetic population, what is the rational to exclude them in this study?
Excellent question, we added an answer in paragraph 4.4. (it is to ensure GLP-1 RA was not prescribed for weight loss).
- Line 100 There is no information on Diabetes medication in the none GLP1 treatment group at all, this is crucial information
This is an excellent remark! We re-ran the study to showcase other anti-diabetic medications. Added to table 1 and to Paragraph 3.
- Line 103 is the PSM list complete? Is only says it includes the parameters given.
That is correct. We are stating what our PSM components are and showcased all components before and after match in Table 1 under outcomes.
- Line 105 choice of gene variants in not explained or justified anywhere and not mentioned once in the rest of the manuscript.
Excellent remark, we added our rationale for inclusion under paragraph 4.3. and 4.4.
- Table 1, BMI should be included and status on the major risk factors for the 2 cancers included in the characteristics. I truly hope that tobacco use has been included in the propensity scoring, same goes for the other major risk factors. If not the entire study is flawed.
We excluded tobacco use and alcohol use along with other risk factors to avoid confounding and to have a clearer association between drugs of study and our outcome. This explanation is also added to paragraph 4.4.
- Table 1, again, no rational for exclusion of above 25 BMI is given.
Addressed previously.
- Diabetes is a major risk factor for a long list of diseases. GLP1 is one of the most effective medications for diabetes and it is of interest if GLP1 treatment beneficial or detrimental for cancer outcome or if the effects seen are due to differences in diabetes management eg is there data in HbA1c available?. However, diabetes medication is not even included in the characteristics and this subject is not mentioned in the introduction or discussion.
This is an excellent remark! We reran the study to showcase the anti-diabetic medications they were on as well as hemoglobin A1C level before and after matching.
- Large part of section 4.1 lacks context with the data presented in should be shortened substantially
Thanks for the comment. We removed non-essential parts. It is aimed at showcasing benefits and side effects of GLP-1 RAs.
- Line 314, why are these genes not included then in propensity scoring and the ones included are not mentioned anywhere?
Great question, we explained our included genes in paragraph 4.4. and removed non-essential ones.
- Line 348-350, again why the BMI 25 cut off?
Addressed previously.
- Line 388, this is extremely vague, instead of lengthy general description of the GLP1/cancer field this should be elaborated and supported by observations and discussed conclusively.
This is a great remark. We expanded the comparison section of our study to the other available ones.
- Line 395, this is not true, the list of variables is not extensive at all, major risk factors are like tobacco and alcohol use, BMI are not stated anywhere or choice of cutoffs not explained.
This is an excellent remark. We excluded tobacco and alcohol use from the study, and explained why we used the BMI cutoff in paragraph 4.4. We changed the phrasing as well. We also added a line to our limitation.
Thank you so much for the critical feedback, such input allows us to improve our manuscript, sets high standards to follow, and ensures quality paper.
Regards,
Reviewer 3 Report
Comments and Suggestions for Authors
Dear author, thank you for sharing this research.
The research investigates whether the use of glucagon-like peptide-1 receptor agonists (GLP-1 RAs) in patients with type 2 diabetes (T2DM) affects the risk of developing gastric and esophageal cancers.
The study examines the safety profile of GLP-1 RAs, particularly concerning cancer risks, and evaluates whether these drugs offer protection against such malignancies.
The innovative aspect of this study lies in the analysis of the long-term cancer risks associated with GLP-1 RAs in T2DM patients, an area that has been insufficiently explored. Additionally, it explores the potential protective effects of GLP-1 RAs beyond their role in diabetes treatment.
The results of a large cohort study suggest that GLP-1 RAs may reduce the risk of gastric and esophageal cancers in T2DM patients, providing new insights into their oncological effects, which could influence clinical decision-making in the management of T2DM.
Suggestions for improving the methodology include:
- Expanding inclusion criteria to account for greater genetic diversity and comorbidities.
- Adding controls for other cancer risk factors, such as smoking, alcohol use, diet, and physical activity.
Regards
Author Response
Thank you for your time and feedback. We appreciate the effort and comments.
Dear author, thank you for sharing this research.
The research investigates whether the use of glucagon-like peptide-1 receptor agonists (GLP-1 RAs) in patients with type 2 diabetes (T2DM) affects the risk of developing gastric and esophageal cancers.
The study examines the safety profile of GLP-1 RAs, particularly concerning cancer risks, and evaluates whether these drugs offer protection against such malignancies.
The innovative aspect of this study lies in the analysis of the long-term cancer risks associated with GLP-1 RAs in T2DM patients, an area that has been insufficiently explored. Additionally, it explores the potential protective effects of GLP-1 RAs beyond their role in diabetes treatment.
The results of a large cohort study suggest that GLP-1 RAs may reduce the risk of gastric and esophageal cancers in T2DM patients, providing new insights into their oncological effects, which could influence clinical decision-making in the management of T2DM.
Suggestions for improving the methodology include:
- Expanding inclusion criteria to account for greater genetic diversity and comorbidities.
Thank you for your comment. We revised the inclusion and exclusion part of the analysis and expanded more on our inclusion and exclusion criteria and why we used certain cutoff and those genetic variants.
- Adding controls for other cancer risk factors, such as smoking, alcohol use, diet, and physical activity.
Thank you for your feedback! We also revised the inclusion and exclusion component to reflect those suggestions. We also added to our limitations the inability to account for diet in the study.
Thanks again for the time and effort you put in. We appreciate your feedback and kind comments!
Regards,
Round 2
Reviewer 3 Report
Comments and Suggestions for Authors
Dear author,
I dont have any other request
This is a good work. No changes needed
Best regards
Author Response
Thank you so much for your comments and feedback which helped us enrich and improve our content!
We appreciate your time and effort!